# Do chimpanzees (*Pan troglodytes*) attribute preferences to virtual competitors?

Emilie Rapport Munro[1]*, Matthias Allritz[2], Kenneth Schweller[3], Daniel B. M. Haun[2], Josep Call[1]

1 School of Psychology and Neuroscience, University of St Andrews, St Andrews, Fife, Scotland, 2 Department of Comparative Cultural Psychology, Max Planck Institute for Evolutionary Anthropology, Leipzig, Saxony, Germany, 3 Ape Cognition and Conservation Initiative, Des Moines, Iowa, United States of America

* ecrm1@st-andrews.ac.uk

## Abstract

Many animal species live in multi-level societies regulated by complex patterns of dominance. Avoiding competition with dominant group-mates for resources such as food and mates is an important skill for subordinate individuals in these societies, if they wish to evade harassment and aggression. Chimpanzees (*Pan troglodytes*) are an example of such a species. This study investigated whether chimpanzees could understand the food preferences of their competitors, and make use of this understanding to select non-contested food items. Fifteen chimpanzees were given thorough experience of the differing target preferences of two virtual competitors. In the test, subjects had to select which of the two targets to approach, based on which competitor was present. To choose correctly, they would have to *integrate* the competitors' preferences from across disparate observations, and then *infer*, before the competitor acted, what they would do in a novel situation. We also included a control condition featuring two targets for which subjects had no information about the competitors' potential biases. The chimpanzees rapidly learned to direct their virtual agent to collect the targets, and some responded with vocalizations and hard knocking against the screen when competitors "stole" targets from the agent the subject was guiding. However, statistical analyses showed that, both at the individual and the group level, they did not succeed in selecting the correct target item at above-chance levels. Additionally, there was no significant difference between their performance in the test and control. We identify theoretical and methodological discrepancies that could explain the contrasting results of this and other studies.

## Introduction

The ability to integrate information about other individuals' past behavior and predict their future actions in order to guide one's own choices is an important skill for many

**Data availability statement:** All data files are available at the OSF: https://osf.io/5asxk/?view_only=5f0a125baa3b46fd8254976b12b12955 The software used for this project is available at the OSF: https://osf.io/j76gn/?view_only=9f0ac-be03a534e4ba97c03aa691dabb9.

**Funding:** ERM and JC were funded by a Diverse Intelligences Grant from the Templeton World Charity Foundation. Grant ID: TWCF-2020-20540. TWCF (https://www.templeton world-charity.org/) played no role in study design, data collection, analysis, decision to publish, or preparation of the manuscript.

**Competing interests:** The authors have declared that no competing interests exist.

animal species. It can prove useful for navigating both cooperative and competitive contexts, and for gleaning social information from third-party interactions [1]. Chimpanzees (*Pan troglodytes*) are highly social, living in complex, multi-level societies with dynamic dominance hierarchies, which in the wild are composed of several dozen individuals [2–6]. For subordinate individuals, who risk incurring repercussions such as aggression if they encroach upon dominants' resources, it is beneficial to avoid competing with more-dominant individuals wherever possible.

Past research has shown that chimpanzees are proficient in this regard. For example, Hare et al. [7] conducted an experiment in which a subordinate and a dominant chimpanzee entered the same room from different sides. The room contained two pieces of food: one was visible to both chimpanzees, while the other was visible to the subordinate but screened from the dominant. Subordinate chimpanzees chose to approach the item that was hidden from the dominant's view. This indicates that chimpanzees are capable of taking into account the mental state of *seeing* as it pertains to other individuals, and that they can make use of this understanding to avoid competitive interactions where these are likely to be detrimental. An initial replication attempt failed to find similar evidence of perspective-taking [8], but Bräuer and colleagues later succeeded in replicating the original results, noting that the first replication suffered from a lack of statistical power and from methodological issues [9]. Note that although critics of mentalistic accounts of apparent perspective-taking in chimpanzees have put forward alternative explanations, such as behavior-reading and stimulus generalization, both the Hare and Bräuer studies included experimental conditions that controlled for these explanations. Various other food competition paradigms have accordingly suggested visual-perspective-taking abilities in chimpanzees: they conceal their approaches from competitors [10], intentionally deceive competitors by approaching food indirectly [11], and strategically manipulate competitors' visual access [12].

Relatively less research has been dedicated to investigating whether chimpanzees can understand others' *preferences*, and use this to guide their actions in the same way. To put it another way, can a chimpanzee pay attention to which of two food items a competitor seems to have a preferential liking for, and so select the non-preferred item to avoid direct competition? Moreover, do they comprehend that these preferences are individually specific, such that another competitor can have the opposite preference and should therefore elicit the opposite behavior?

Eckert and colleagues [13] have shed some light on this question. In their study, chimpanzees gained experience of the conflicting food biases of two human experimenters; then, in test trials, the subjects expected the experimenters to make choices in line with their biases (except when they could not see what they were choosing), and the chimpanzees used this expectation to guide their own choice of which experimenter to receive food from. In another study, Buttelmann and colleagues [14] demonstrated that chimpanzees used the valence and directedness of a human experimenter's emotional expressions to infer which of two boxes he had eaten from; the chimpanzees then chose to search for food in the other, untouched box.

Recently, Kaminski and colleagues [15] found contrasting results. In their study, subjects chose between two food items after human competitors with different biases had made a choice in secret, such that the subjects did not know which item was still available. When interacting with a competitor whose preferences matched the subjects', the correct response was to choose their own less preferred food. All four species of great ape failed to do so. This could indicate that the apes were unable to recognize the competitors' preferences; however, another potential explanation is that they were merely unable to inhibit their own desire to select their preferred food.

Considering the conflicting findings of the above studies, further work is required in a variety of contexts to provide robust support or contradictions of the hypothesis that chimpanzees recognize others' preferences. The current study aims to deliver this, using a novel method for simulating social interactions: competing with virtual agents.

Over the past several years, virtual environments have been used with increasing frequency to study great apes' navigational and spatio-cognitive skills [16–21]. Rapport Munro and colleagues [21] recently conducted the first experiment to investigate *social* cognition within the virtual realm. Chimpanzees and bonobos were tasked with "catching" (i.e., guiding their invisible virtual agent to collide with) virtual rabbits that ran away when approached. All subjects learned rapidly how to chase and catch the rabbits, despite only previously having experience with static virtual targets; moreover, generative computational modeling techniques revealed that on a subset of trials, subjects intercepted the rabbits by anticipating their movement trajectories.

The benefit of using virtual environments to study ape social cognition is that they provide a good balance between logistical and ethical considerations on the one hand, and experimental validity on the other hand. Some of the best and most-cited evidence of chimpanzees considering the states of minds of others [7,22] comes from real-life competition experiments involving two conspecifics simultaneously going after the same target in the same room. However, important ethical and welfare advances have rendered such experiments difficult to administer. Using virtual environments, we can return to something akin to those original methods while also controling for potential behavior-reading and prior-experience explanations.

The current study investigated whether chimpanzees could learn the differing prey preferences of two novel virtual competitors, and subsequently use this knowledge to succeed in competitive interactions by choosing to approach the non-preferred prey of whichever competitor was present. To rule out associative learning explanations, we also included a control condition featuring prey items with regards to which subjects had no information about the competitors' potential biases. We hypothesized that if the chimpanzees were responding to the competitors' preferences, they should choose the correct prey at above-chance levels in the Test condition, but not in the Control.

## Experiment 1

### Methods

**Subjects and housing.**  Subjects were six chimpanzees (one male, five females, mean age at start of data collection = 25.8, SD = 16.4) housed at the Wolfgang Köhler Primate Research Centre (WKPRC) in Leipzig Zoo, Germany (see Table 1). All subjects had prior experience with navigating three-dimensional virtual environments via the touchscreen, although the extent of this experience varied from individual to individual.

Research was conducted in a dedicated testing room adjacent to subjects' main indoor enclosure. All participation in the study was voluntary, and subjects were never deprived of food or water. Except for dependent infants, each subject was separated from the rest of their group for the testing period, which never exceeded 30 minutes per subject per day (with a maximum of 24 trials per day). All animal husbandry and scientific research conducted at WKPRC is within the standards set out by the European Association of Zoos and Aquaria (EAZA) and the World Association of Zoos and Aquariums (WAZA). Ethical approval for the study was granted by the University of St Andrews School of Psychology & Neuroscience Ethics Committee (approval code PS16860) and by the head keeper, research coordinator, and director of the WKPRC.

**Table 1. Experiment 1 subject information.**

| Subject | Age at start of testing | Sex | Previous VE experiment experience (beyond training)? |
|---------|------------------------|-----|------------------------------------------------------|
| Azibo | 8 | Male | Yes |
| Fraukje | 47 | Female | Yes |
| Riet | 45 | Female | No |
| Swela | 27 | Female | No |
| Tai | 20 | Female | Yes |
| Youma | 5 | Female | No |

In this table, 'VE experiment' refers to experiments in which subjects used a touchscreen to interact with Virtual Environments and collect virtual rewards.

## Materials

During testing times, a clear infrared touchscreen measuring 27" diagonally was installed in place of one of the panels in the research room, with an equal-sized color computer monitor attached behind it. The monitor, connected to a laptop, displayed the stimuli through the clear touchscreen. Two stereo speakers were also attached to the laptop, through which stimulus sounds were played. During testing, subjects were filmed from behind and above on a camera connected to a tablet that was mounted to the back of the computer monitor, allowing the researcher to view the video feed in real-time. The software used was APExplorer3D, programmed in C# using Unity3D. The virtual environment was presented from the first-person perspective of an invisible agent. When subjects touched a location on the screen other than the sky, the invisible agent would walk to that location, unless it contacted an object it couldn't walk through, or another touch was recorded. An exception to this was that if subjects touched the very right- or left-hand edges of the screen, the agent would turn on the spot.

**Stimuli.** The target objects were four prey items of similar size: a rabbit, a dog, an antelope, and a chicken (Fig 1A). Subjects could approach the prey by touching them or a location near them. When subjects approached within two virtual meters of the prey, the prey would be triggered to walk away along a predetermined path. This was done to convey to the subjects that virtual prey would behave similarly to real-life agentive, living prey, while at the same time making the prey not too difficult to "catch". For this reason, the evasive path of the prey was very short, and once they had completed this path, they remained stationary, meaning that they could be caught if the subject followed them. When subjects made contact with a prey item, the prey would fall over, and the speakers emitted a "ta-daah" sound to indicate that the trial had been successful.

The competitors were two animals, significantly larger than the prey items: a gorilla and an elephant (Fig 1B). As with the prey targets, the competitors could walk around the virtual environment in a realistic-looking manner; in addition, to add to their saliency, the competitors would vocalize at various points in the trial. Both the gorilla and the elephant competitors made similar chimpanzee food grunt vocalizations, recorded from different individuals [23]. Both individuals whose grunts were used were unfamiliar to the subjects of this study.

**Rewards.** Each time a subject succeeded in contacting a prey item, the experimenter rewarded them with a small piece of apple *and* either a slice of banana or half a grape (alternating across trials). Different prey items were not rewarded differentially.

## Procedure

The experiment began with two distinct training stages, followed by the testing stage. Please see Fig 2 for a graphical representation of the procedure.

**Training 1.** Subjects were introduced to the four aforementioned prey. Some trials presented each prey item separately, while other trials featured all four prey together. At the start of each trial, prey appeared in one of four different

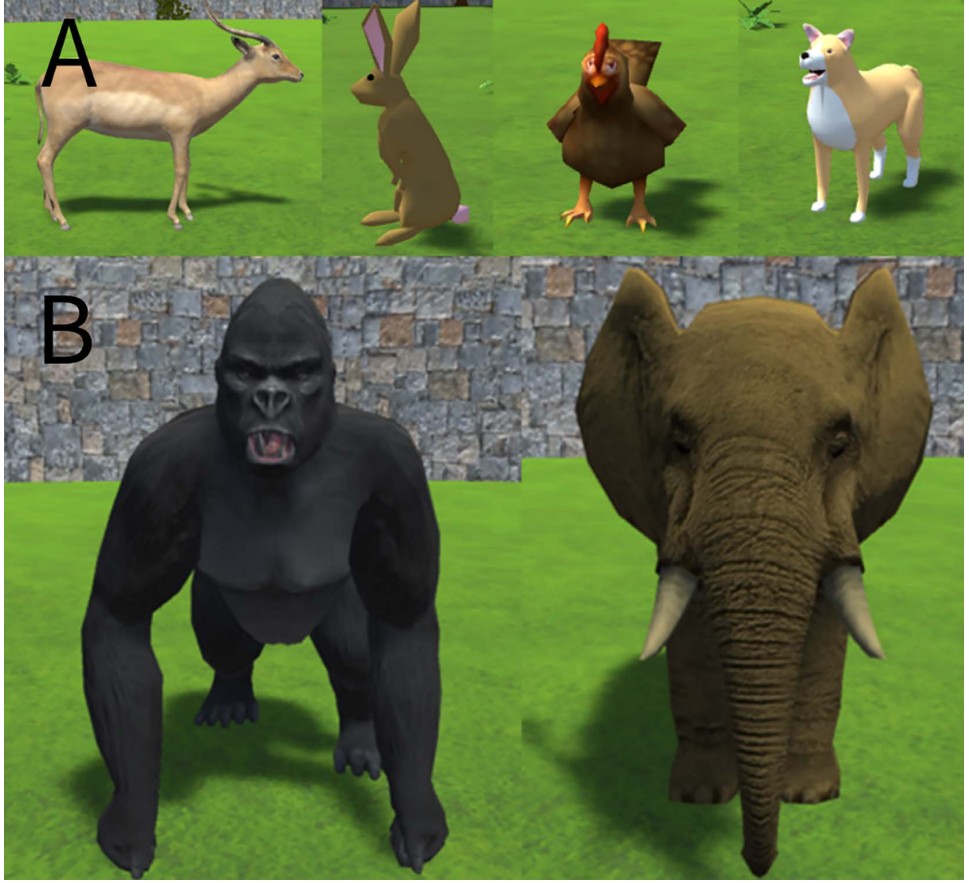

**Fig 1. Virtual animals.** This figure shows the virtual agents with which subjects interacted in the virtual environment. A: the virtual prey targets (from left–right: antelope, rabbit, chicken, dog). B: the virtual competitors (from left–right: gorilla, elephant).

locations in the virtual environment, always within sight of the subject's viewpoint. Subjects were rewarded each time they collected a prey item, such that four-prey trials promised up to four rewards. One-prey trials lasted up to a maximum of 50 seconds, and four-prey trials lasted up to 120 seconds; if a trial timed out without the subject having caught one or more of the prey, the subject would lose out on the food reward associated with that target.

The aim of this first stage of training was to habituate the subjects to these four prey types, and to teach them that although they were different in shape and size and displayed subtly different movement patterns, they all behaved identically, were equally easy to catch, and resulted in equal rewards. Each subject received one session of this training per testing day; each session was made up of 15 trials (three of each prey type alone and three with all prey together). All subjects received a minimum of five sessions of this training. If they collected all prey items in at least 14 out of 15 trials in the fifth session, they moved on to the next training stage; if they failed to do so, they continued with training stage 1 until they reached that criterion, *or* until they collected all prey in 13 out of 15 trials in two consecutive sessions. No subject required more than the minimum five sessions; however, one subject, Swela, on the first training day showed difficulty understanding that the animal prey were the targets that should be approached. The experimenter therefore aided her in several trials by taking over control of the agent via keyboard input, orientating the agent and guiding it towards the prey, to convey to the subject that the agent contacting the virtual prey would result in reward. This was only required for a small number of trials, after which she began to approach the prey of her own accord. Due to this intervention, the first day of

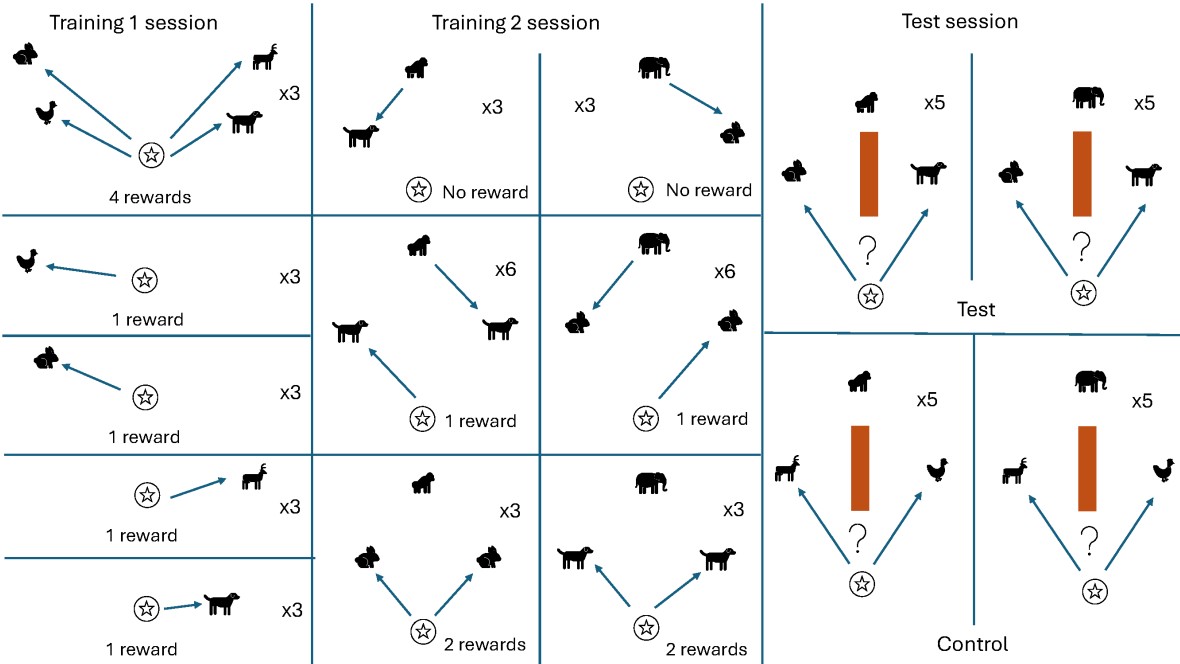

**Fig 2. Graphical representation of experiment 1 procedure.** This figure shows the different types of trial present in each of the three phases of the experiment: Training 1, Training 2, and Test. The player is represented by the star; the competitors are represented by the gorilla and the elephant; the prey are represented by the dog, rabbit, antelope, and chicken. Each panel shows a different trial type; for each one, it is indicated a) how many times such a trial appeared within a session, and b) how many rewards a subject would receive in this trial (with the exception of trials in the Test phase, since whether or not a subject was rewarded depended on their actions in these trials).

training with Swela was disregarded, and her five sessions of training were counted from the following day (from which point no further experimenter intervention was given).

This was the first time that any of the subjects had experienced anything in the virtual world that resembled animals, as well as the first time they experienced targets that moved away when approached.

**Training 2.** In this stage, we introduced the gorilla and elephant competitors. The only prey items present in this stage were the dog and the rabbit; the chicken and antelope were reserved for the test. The aim of this stage was to teach the subjects about the behavior of the competitors, in particular with respect to two facets: their preferences and their speed. The gorilla only chased after the dog as its prey, and ignored the rabbit; conversely, the elephant would only chase the rabbit and ignore the dog. Additionally, trials were designed so that in *every* instance in which a subject attempted to catch the same prey item as one of the competitors, they would always lose out because the competitor would get there first. Trials in this stage took the following three formats:

*Gorilla Catches the Only Dog/Elephant Catches the Only Rabbit.* These trials featured one of the two competitors and the prey item preferred by that competitor. At the outset of a trial, the competitor began directly opposite the subject avatar, some distance away. Between the competitor and the subject, and off to either the left or the right, was the prey. The prey was always significantly closer to the competitor than to the subject (see Fig 3A).

As soon as the trial started, the competitor vocalized. In response to the competitor's vocalization, the prey jumped up a small distance into the air to convey that they recognized the predator as a threat. Three seconds after the trial began, the competitor began to approach the prey, regardless of what the subject did. Once the competitor entered within 1.5 virtual meters of the prey, the prey took a short course of evasive action, similarly to Training stage 1, culminating in the competitor catching the prey.

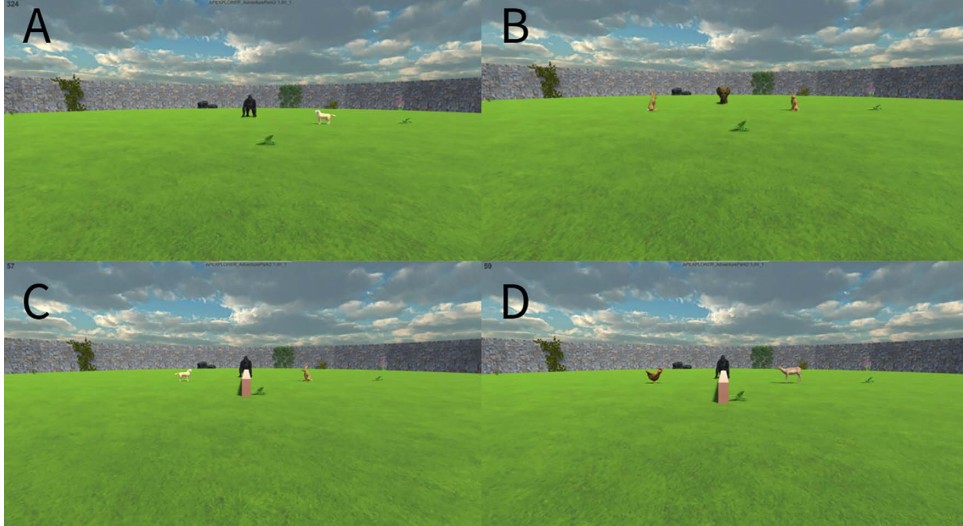

**Fig 3. Start screens in various trial types.** This figure shows the image that subjects would be faced with upon beginning trials of different types. A: example start screen in a 'Gorilla catches dog' trial in training stage 2. B: example start screen in an 'Elephant Catches One of Two Rabbits' trial in training stage 2. C: Example start screen for a gorilla test trial. D: Example start screen for a gorilla control trial.

When this happened, the prey fell over as in Training 1, and the competitor made a higher-frequency version of the food grunt they made at the start of the trial. The competitor also made a celebration gesture: the gorilla stood up on its back legs and beat its chest, and the elephant reared itself up and raised its trunk. The trial then ended. The reason for adding these additional gestural and vocal cues was that we reasoned that a more emotionally evocative scene would be more memorable, thereby helping with the learning that would be required in order for the subjects to integrate their experiences in the training period into an understanding of the competitors' preferences.

These trials each lasted approximately 14 seconds. The subject was not able to direct their avatar to move for the first two seconds of the trial—no matter where they touched on the screen, the avatar would not move. After these two seconds elapsed, the subject could "walk" through the environment towards the prey. However, due to the relative positions of the three agents, and the fact that the competitors were programmed to move more swiftly than the subject's avatar, the subject always lost to the competitor in these trials, no matter whether or not they attempted to collect the prey.

*Gorilla Catches One of Two Dogs/Elephant Catches One of Two Rabbits*. It was desirable to preclude the possibility of subjects forming an association between the competitors and prey such as, 'SEE DOG + SEE GORILLA = FAILURE, therefore when SEE DOG + SEE GORILLA, AVOID DOG'. This is because such learned cues would enable them to succeed in the test stage without requiring any understanding of the relevant behavior of the competitors. We therefore included trials featuring one competitor and *two* of their preferred prey (Fig 3B). The competitor would take one and the subject could "collect" the other, so they would still get the experience of winning a reward in a situation containing that specific prey-competitor combination.

Three seconds after a trial was initiated, the competitor would begin moving towards one of the prey items (in half of the trials the competitor chose the prey on the right, and on the other half the left). After collecting the prey, it remained stationary. The subject's avatar was restricted from moving for the first two seconds of the trial; following this, they were free to "collect" the prey item that the competitor did not go for. These trials lasted a maximum of 90 seconds, but would end as soon as both prey were "collected", meaning that they were usually much shorter than the maximum length.

*Gorilla Ignores Two Rabbits/Elephant Ignores Two Dogs.* To communicate the idea of the competitors' preferences, we presented predators as they ignored their non-preferred prey, as well as chasing their preferred prey. In addition, it was important to balance the number of times a subject could catch each type of prey in the presence of each competitor, to preclude the same associative rule described above. Trials of this third type therefore featured one competitor and *two* of their *non-preferred* prey. The competitor stayed silent and stationary throughout, and the subject had 90 seconds to collect both prey items.

Each subject received nine sessions of this training, with a maximum of one session per testing day. Each session consisted of 24 trials, allotted to each of the trial types as shown in Table 2:

The counterbalancing ensures that across trials within a session, subjects catch an equal amount of dogs and rabbits (6 each) while in the presence of each competitor, while still getting experience in a competitive context regarding which prey is preferred by which predator. This arrangement meant that in a full session, across the four competitor/prey combinations, there was perfect balance both in the *absolute number of rewards* a subject could receive, and the difference of success/failure, calculated as *number of successful trials* (in which they got at least 1 reward) *minus the number of unsuccessful trials* (see Table 3).

**Test.** Here, for the first time, each competitor was presented in conjunction with both the dog and the rabbit. As in the training, the competitor began opposite the subject, with one prey item to the left and one to the right, again closer to the competitor than to the subject. In this stage, we also included a low brick wall running parallel between the subject and the competitor (Fig 3C). This ensured that once their approach to the prey item had begun, the subject could not change their mind and collect the other target.

In these trials, the competitor vocalized to indicate that their preferred prey was present, but did not move towards either of the potential targets. To avoid giving away which of the prey would be afraid of the competitor in question, the prey did not jump in response to the competitor's vocalization as they did in Training 2. The subject had the choice of approaching either the dog or the rabbit. If they chose the correct prey—the rabbit on gorilla trials, and the dog on elephant trials—the competitor would remain stationary and the subject would be able to collect the prey and receive the reward. If, on the other hand, they approached the competitor's preferred prey, they crossed an invisible trigger line that

**Table 2. Session structure of experiment 1, training stage 2.**

| Trial type | Gorilla catches dog | Elephant catches rabbit | Gorilla catches 1 dog | Elephant catches 1 rabbit | Gorilla ignores 2 rabbits | Elephant ignores 2 dogs |
|---|---|---|---|---|---|---|
| Number of trials in a session | 3 trials | 3 trials | 6 trials | 6 trials | 3 trials | 3 trials |
| Subject's competitor | Gorilla | Elephant | Gorilla | Elephant | Gorilla | Elephant |
| Subject catches | Nothing | Nothing | 1 dog | 1 rabbit | 2 rabbits | 2 dogs |

This table indicates the number of trials that were allotted to each of the six trial types in a single session of stage 2 of training.

**Table 3. Reward schedule structure of experiment 1, training stage 2.**

| | Trials-with-reward *minus* trials-without-reward | Number of rewards |
|---|---|---|
| Gorilla in presence of dog(s) | 3 | 6 |
| Gorilla in presence of rabbits | 3 | 6 |
| Elephant in presence of rabbit(s) | 3 | 6 |
| Elephant in presence of dogs | 3 | 6 |

This table indicates the number of rewards, and the ratio of rewarded to non-rewarded trials, a subject would receive across each of the four combinations of competitor and prey. Balancing the reward schedule in this way should rule out success by associative learning.

caused the competitor likewise to move towards the prey. As before, the competitor was programmed to always reach their prey before the subject could get to it, upon which the competitor would vocalize and celebrate as in the training, and the trial would end without the subject being rewarded. Trials lasted a maximum of 90 seconds, and the subject could direct their avatar to move as soon as the trial began.

In addition to the Test condition, we included a Control condition featuring the chicken and the antelope prey from the first stage of training (Fig 3D). The inclusion of the Control was to account for the possibility that success in the Test condition could be explained by subjects learning the reward contingencies over the course of the test, as opposed to bringing forward an understanding of the competitors' likes and dislikes from the training.

As with the rabbit and dog, the two competitors had strong and opposite preferences for these prey (the gorilla would chase the antelope, and the elephant would chase the chicken), but subjects had no experience of these preferences. If subjects showed a preference for the rabbit in test trials with a gorilla (and vice-versa), earlier than they learned to choose the chicken in the presence of the gorilla (etc.), this would point to their using their experience of the gorilla's and elephant's preferences from the training.

Test sessions comprised 20 trials: 10 test and 10 control, with an equal number of gorilla and elephant trials in each. Likewise, each prey featured on the left-hand and right-hand side of the arena an equal number of times. The order of trials within a session was randomized, but always presented in blocks of five test trials and five control trials. Subjects each received 10 test sessions, with a maximum of one session per testing day. One exception was Youma, whom we discontinued testing partway through her ninth session, due to her exhibiting signs of distress, specifically fear-grinning and banging excessively hard on the touchscreen after a succession of trials in which she lost out on the rewards to the virtual competitors. Since she had so nearly completed the full complement of test sessions, we elected to include her in the analysis.

Please see Supporting S1 Video for a clip of subject Azibo completing three test trials.

## Analysis

We conducted separate analyses for each subject, and also grouped subjects' data together and analyzed their behavior as a whole.

First, we wished to ascertain whether the observed success levels differed from those expected by chance. We set the chance level at 0.5, as is standard for binomial tests in which under the null hypothesis (i.e., randomness) the expected probability of success on any one trial is 0.5 [24]. In our study, there were two choice options (two prey items) in every trial, and "success" was denoted as where a subject chose correctly, i.e., chose the non-preferred prey of the competitor present, while "failure" was where they chose incorrectly. We anticipated, when planning the test, that the percentage of trials that passed without one of these two choices being made would be negligible based on the subjects' high success rate in the training stages in which prey could be caught. This informed our decision to set the chance level of 0.5. Indeed, there were zero instances of a chimpanzee not choosing one of those options (i.e., choosing not to "move" at all within the virtual environment, or "walking" in a direction not towards either of the prey). We conducted separate binomial tests for the Test condition and the Control condition.

Second, we compared performance between Test and Control conditions. We assessed the normality of data using a Shapiro test. If the normality assumption was not violated, we used a parametric paired samples t-test, where "session" served as the grouping variable. The reason for using paired samples t-tests was that the samples were non-independent due to the within-subjects design. If the data were non-normal, we used a Wilcoxon signed-rank test instead.

All analysis was conducted in R version 4.1.2 [25]. Base R was used, with no additional packages employed.

## Results

Success levels for each subject across both conditions in the test are shown in Fig 4 (see S1 Table in the Supporting Information for individual binomial test results).

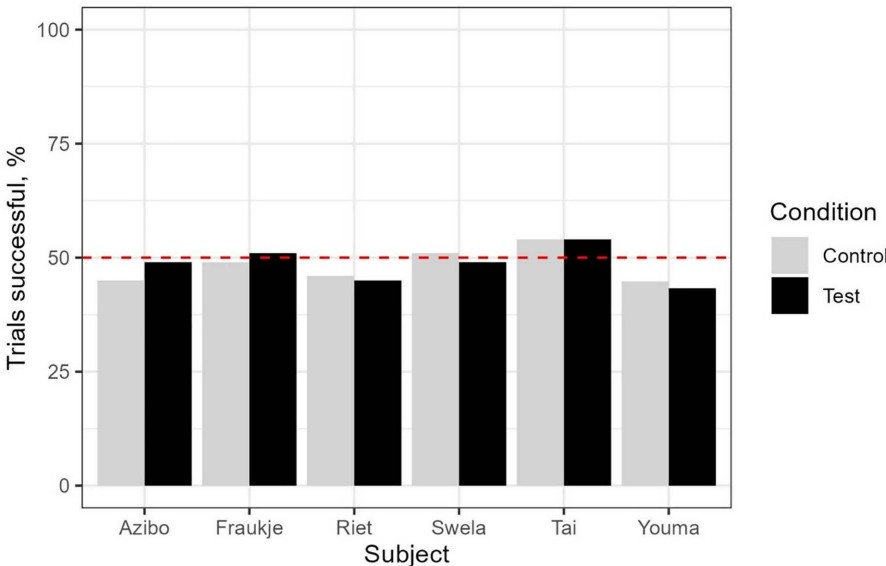

**Fig 4. Individual success rates across the control and test conditions in experiment 1.** This figure indicates, for each subject, the percentage of trials in which they succeeded in both the Test and the Control conditions. The dotted red line shows the chance level of 50%.

For all subjects, in both the Test and Control conditions, performance was not statistically different to chance levels (binomial tests, see Supporting S1 Table).

We further compared the success levels between the Control and Test conditions, and found that subjects as a group showed no significant differences across the two conditions (paired t-test, t(5)=−0.264, p = .802). Analysis of each subject's performance individually returned similarly non-significant results (see Table 4).

We noted during the data collection period that many subjects appeared to show strong side biases, seeming to prefer approaching the prey on either the left side or the right, irrespective of the presence of competitors or the relative positions of the prey. We quantified these biases using binomial tests and found that all six subjects chose one side over the other significantly more often than would be expected by chance. Three subjects favored the prey on the right-hand side, and three favored the left. Consult Supporting S2 Table for the percentage of trials each subject went to each side in the two conditions and the results of the binomial tests. Additionally, it was noted that some subjects seemed to have a preference

**Table 4. T-tests comparing individual and whole-group success rates across the control and test conditions.**

| Subject | Test statistic | Degrees of freedom | P |
|---|---|---|---|
| Azibo | t = −0.802 | 9 | .443 |
| Fraukje | t = −0.361 | 9 | .726 |
| Riet | t = 0.218 | 9 | .832 |
| Swela | V = 1 *see note | 9 | 1 |
| Tai | t = 0 | 9 | 1 |
| Youma | t = 1.323 | 9 | .228 |
| All subjects | t = −0.264 | 5 | .802 |

This table indicates the results of t-tests that were used to identify, at both individual and group levels, whether success rates differed significantly between the Test and Control conditions. The data from subject Swela failed to meet normality expectations, so these data were analyzed using a Wilcoxon signed-rank test rather than a parametric t-test.

for one particular prey item over the other, despite the fact that all prey should have been equal for them. Using binomial tests, we found that two subjects showed a significant preference for the chicken over the stag in Control trials; in Test trials, one subject showed a significant preference for the rabbit over the dog (see S3 Table in the Supporting Information).

## Discussion

We found no evidence for chimpanzee subjects using their prior experience of their competitors' prey preferences to inform their own choices in a two-option test. In Test trials, subjects' performance did not differ from that expected by chance. Furthermore, there was no statistical difference between their performance on Test and Control trials, either individually or as a group. This suggests that the experience of the competitors' behavior patterns they had been afforded in stage 2 of Training had no effect on their choices in the test. How do we explain this failure to make use of experience?

One possibility is that the training we gave the chimpanzees was insufficient. For example, perhaps the behavior of the competitors and the prey in the training trials was not salient enough to carry across the message of the gorilla's and elephant's preferences. We considered the saliency of the stimuli in the design stage, adding both visual and auditory components, but it is possible nevertheless that the stimulus presentation was not sufficiently interesting to fully capture the subjects' attention. To this point we would, however, add that subjects were almost always attentive to the screen while a trial was ongoing, showing that they were at least watching what was going on (although but it was fairly common that between trials, subjects might move away from the screen and walk around the testing room, or respond to vocalizations from other members of the group that could sometimes be heard from other parts of the building).

It is also conceivable that the observational information that we offered alone is not enough to promote reasoning about other agents' preferences. In the real-world circumstances (as opposed to experiments using virtual environments) in which they would be using the skill in question (such as competing with group-mates for mating opportunities, or access to food or shelter), chimpanzees would have the opportunity to interact with their competitors in a variety of ways, including through gestural and vocal communication—and this could lend increased strength and saliency to their mental conceptions of their competitors' preferences. However, a number of social learning experiments have shown that chimpanzees can indeed benefit from observational information even when it is divorced from social interaction. For example, Tennie et al. [26] used the 'floating peanut task', in which chimpanzee subjects are required to spit water into a narrow tube until the water level is high enough to bring the floating peanut into reach. Chimpanzees in this experiment were able to learn this skill when it was demonstrated by a conspecific; they were also equally successful when the only demonstration they received was a human experimenter pouring water into the tube from a bottle. As the human experimenter never directly interacted with the subjects, these results show learning and emulation from observation alone. In a related experiment, Marno et al. [27] tested whether the presence or absence of ostensive cues would affect chimpanzees', bonobos' and orangutans' propensity to socially learn to operate a food dispenser. The apes succeeded in operating the device at above-chance levels both when the human demonstrator ostensively communicated with them before carrying out the demonstration, and when there was no communication; there was no difference in success between these two conditions, showing that the apes learned the skill equally well when reliant on observation alone.

However, these studies investigated chimpanzees' ability to learn skills, rather than information about other individuals, as is the case in the current experiment. A closer analogue was conducted by Keupp and Herrmann [28]: chimpanzees were shown two different conspecifics interacting with the same puzzle box, and either succeeding or failing to extract food from it. When given the choice to recruit one of these individuals to be a cooperative partner in operating the puzzle box, the subjects immediately and spontaneously selected the skilled partner. However, when the task was changed to a competitive scenario, such that subjects would benefit from recruiting the less-skilled partner, they failed to do so immediately following the demonstrations, instead requiring experience of working with both individuals before they showed a preference for the unskilled partner. In this competitive scenario, therefore, chimpanzees failed to make strategic decisions based on inferring other individuals' behavior from previous observations, mirroring the current results. Keupp and

Herrmann [28] suggest that their pattern of results could be attributed to the increased cognitive demands of the competitive task, as it requires the subjects to take into account that their own goals and those of their competitor are in conflict. If we accept this explanation, we could suggest that while observational information is sufficient for chimpanzee learning in some contexts, such as skill acquisition (as demonstrated in [26,27]), it may be inadequate to inform optimal decision-making in other scenarios that call for complex perspective-taking [28].

Moreover, in the real world, chimpanzees would be motivated to avoid competing with dominant conspecifics not only because doing so would result in their losing out on the available rewards, but also because it could cause a physical altercation. In our experiments, choosing the incorrect prey did not carry such a risk, and so may not have proved motivating enough for the subjects to attempt to avoid competition.

Another potential explanation is that our subjects did not view the gorilla and elephant as agents, and therefore did not expect them to have stable goals that could be explained by food preferences [29–32]. We made some observations over the course of this Experiment which have some bearing on this point. Throughout stage 2 of Training and the Test, in some instances in which they lost out on the prey to the gorilla or the elephant, some of our subjects made interesting responses that seemed directed towards the competitors themselves. These responses included vocalizations such as grunts; spitting at the touchscreen; and poking the touchscreen or banging it with their fists, often directly on the image of the virtual competitor. It is possible that these represent pure frustration responses to losing trials, but our subjects had previously had experience of losing out on some rewards in touchscreen tasks, and had rarely shown any response of the like towards stimuli presented on the screen. We believe it is conceivable that these responses are indicative of these subjects perceiving the gorilla and elephant as intentional agents. Future experiments could investigate this possibility by replicating the current study's virtual competition scenario with the addition of new competitors that lack agency cues—for example, being simple colored shapes rather than resembling animals, or being stationary rather than self-propelled—and testing whether chimpanzees show more frequent and intense emotional responses towards the agentive than the non-agentive competitors.

Another potential explanation for chimpanzees' failure in this test is that they do possess the ability in question, but some aspect of this test hindered their capacity to deploy it. Perhaps the presentation of the prey items on the left and right of the screen was an issue: as well as asking subjects to choose between two prey options, we were also asking them to pick a side to walk to, and these two choice modalities may have interfered with one another. Handedness, or behavioral lateralization, is prevalent among chimpanzees; although it remains debated whether they display population-level handedness as humans do [33], it is well established that individual chimpanzees often have preferred hands to use for manual tasks, although this preference can vary between tasks [34–37]. Furthermore, it has been shown that captive chimpanzees' motor skill performance is improved when using their dominant hand as opposed to their non-dominant one [38,39]. This individual laterality may explain why all subjects in this experiment displayed a significant side bias. Perhaps, depending on whether an individual prefers to use their left or right hand to operate the touchscreen, it required less effort to approach the prey on one side rather than the other.

Some subjects also showed biases towards specific prey; the only obvious explanation here is that these subjects, for whatever reason, liked the look of some of the prey items better than others.

It is possible that our subjects *were* aware of the contingencies associated with the competitor-prey combinations, but that their side preferences or prey preferences were strong enough to cause them to disregard this information in favor of continually choosing the prey on their preferred side/the prey that they liked the look of better. This may be a valid stratagem: it is far less cognitively taxing than choosing which side to approach on a trial-by-trial basis depending on which competitor is present and the relative locations of the two prey items, and it results in subjects being rewarded 50% of the time, which may be sufficient to keep them satisfied.

However, another possibility is that both types of biases are simply results of uncertainty. Not knowing which prey to choose in any given trial, due to a failure to incorporate any knowledge about competitors' preferences, perhaps the

chimpanzees fell back on preferences related to the side it was easiest for them to approach based on their handedness, or the prey they preferred, as opposed to choosing randomly.

Finally, it's possible that our decision to balance the reward schedule in terms of number of successful trials minus number of failed trials rendered the test too difficult for the subjects to pass. Even if they were considering the competitors' preferences and acting accordingly, this reward schedule asks them to disregard the fact that in terms of *absolute numbers of trials*, they are rewarded *more* frequently in the 'incorrect' pairings (gorilla/dog and elephant/rabbit). This could have interfered with subjects' parsing of the reward contingencies and their abilities to make the necessary associations between each competitor and its preferred prey. We addressed this possibility in Experiment 2, as well as aiming to increase the saliency of the training. In other words, whereas in Experiment 1 we tried to avoid confounding predator-prey combinations with reward frequency, in Experiment 2 we deliberately confounded them to give subjects the best opportunity to learn about the virtual agents' likely future behavior.

## Experiment 2

### Methods

**Subjects and housing.** Subjects were nine chimpanzees (one male, eight females, mean age at start of data collection = 28.1, SD = 16.2) housed at the WKPRC who had not participated in Experiment 1 (see Table 5). Three of these subjects had just completed their training on touchscreen use, and this was their first experience of an experiment in virtual reality. The other subjects had completed previous experiments within this virtual world, but the extent of their experience varied.

### Materials

All equipment (including the touchscreen, computer and camera used) was the same as in Experiment 1. The stimuli (the virtual environment, virtual competitors, and virtual prey) were also the same as in Experiment 1, except that we used a virtual boar in place of the virtual dog prey (see Training 1, below).

### Procedure

Please see Fig 5 for a graphical representation of the procedure in Experiment 2.

**Training 1.** This was identical to Experiment 1, except that the dog prey was replaced by a black boar (see Fig 6). We had determined that the coloration of the dog was perhaps too similar to that of the rabbit, and that saliency could be increased by choosing a different animal. Subjects received five sessions, one per testing day.

**Table 5. Experiment 2 subject information.**

| Subject | Age at start of testing | Sex | Previous VE experiment experience (beyond training)? |
|---|---|---|---|
| Alex | 22 | Male | Yes |
| Carola | 2 | Female | No |
| Changa | 12 | Female | No |
| Corrie | 46 | Female | No |
| Daza | 37 | Female | No |
| Frederike | 49 | Female | Yes |
| Hope | 32 | Female | No |
| Sandra | 29 | Female | Yes |
| Zira | 25 | Female | No |

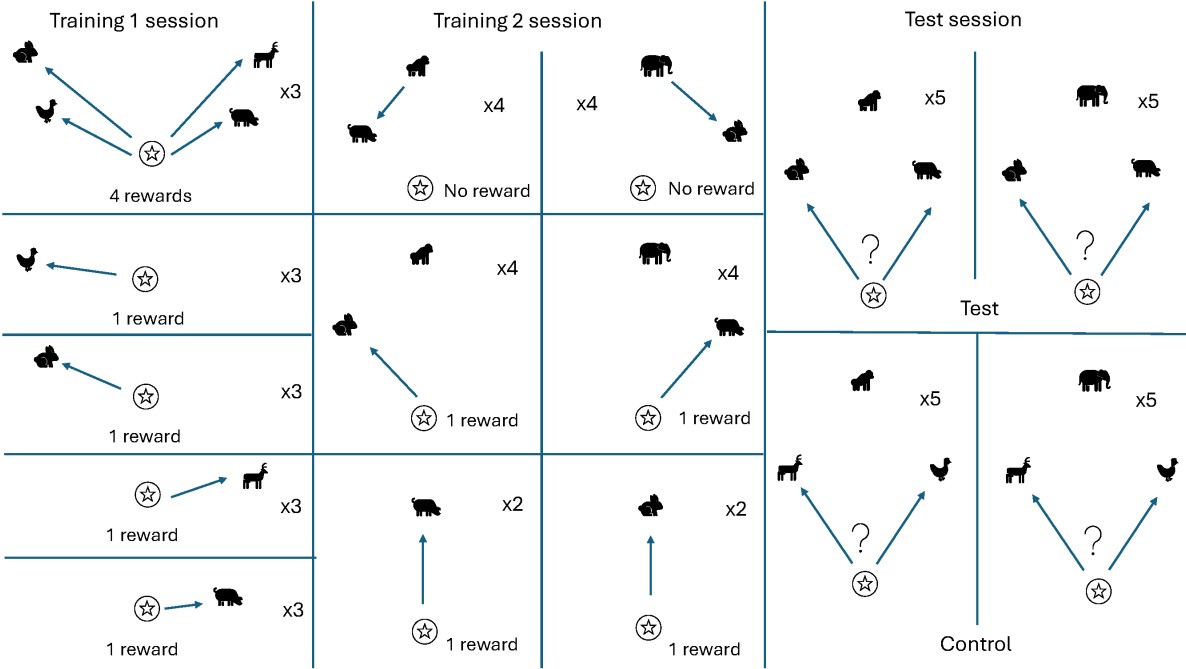

**Fig 5. Graphical representation of Experiment 2 procedure.** This figure shows the different types of trial present in each of the three phases of the experiment: Training 1, Training 2, and Test. The player is represented by the star; the competitors are represented by the gorilla and the elephant; the prey are represented by the boar, rabbit, antelope, and chicken. Each panel shows a different trial type; for each one, it is indicated a) how many times such a trial appeared within a session, and b) how many rewards a subject would receive in this trial (with the exception of trials in the Test phase, since whether or not a subject was rewarded depended on their actions in these trials).

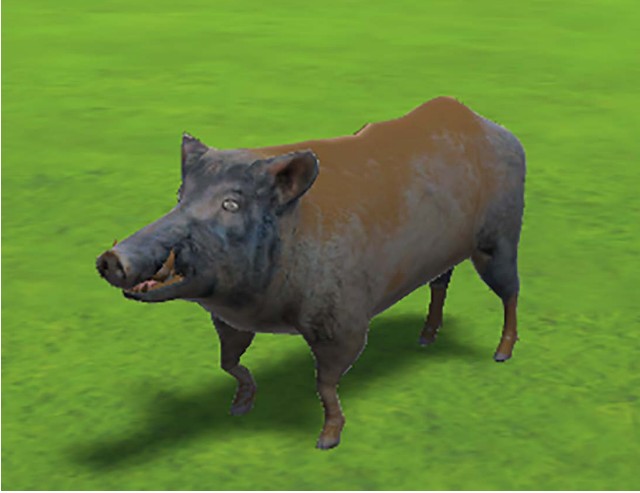

**Fig 6. Boar target.** This figure shows the virtual boar that replaced the virtual dog as a prey item in Experiment 2.

**Training 2.** The setup of trials in this stage of training was the same as in Experiment 1, except that the competitors and prey items were stationed closer to the subject than they had been originally. This change was made to render the competitors and prey more easily visible and thereby potentially more salient to the subjects. Subjects each received eight sessions of this training, one per testing day. Each session contained 20 trials which fell into three categories:

*Gorilla Catches the Only Boar/Elephant Catches the Only Rabbit.* These trials functioned just as they did in Experiment 1: the competitor vocalized and then chased and caught the prey, celebrating after doing so, and the subject was left with nothing. Each session contained four of each.

*Gorilla Ignores the Only Rabbit/Elephant Ignores the Only Boar.* Unlike in Experiment 1, these trials contained only *one* of the competitor's non-preferred prey, but in all other respects they were identical: the competitor remained still and silent, and the subject was free to collect the prey. Each session featured four of each trial.

*Rabbit/Boar with no competitor.* We also included trials from stage 1 of training, featuring just one prey item and no competitor, to remind subjects that they could collect either prey, and also to function as 'easy' trials to provide rewards and maintain motivation. Each session contained two rabbit trials and two boar trials.

**Test.** Trials in both the Test and Control conditions were very similar to those in Experiment 1 except for the following details: as in training, the competitors and prey were moved closer to the subject's starting point. Additionally, we omitted the brick wall separating the two prey items, due to the observation that subjects in Experiment 1 had occasionally become stuck on it when trying to approach their chosen prey, and also because it had become clear that it was not required to prevent subjects from changing their choices after having chosen a target. Once a subject had begun to approach the incorrect prey item, the speed at which the competitor moved to collect that prey meant that even if the subject then changed their choice and tried to catch the other prey, they did not have time to do so before the competitor caught its prey and the trial ended.

Notice that in this experiment, we omitted training trials that featured two of the competitor's preferred prey, where the competitor would collect one prey and the subject would be able to collect the other. This was done to address the possibility that such trials may have confused the subjects in Experiment 1 or interfered with their ability to attribute preferences to the competitors, since the competitors may have seemed to ignore prey items that they were supposed to like. However, this change means that there were fairly simple associative rules that subjects could learn throughout the course of training that would enable them to pass the test stage. They would always lose out on the rabbit in the presence of the elephant and would lose the boar in the presence of the gorilla, and if they noticed these contingencies they could use them to avoid the corresponding prey in test trials without needing to understand the *reason* behind these contingencies. Therefore, if the chimpanzees pass this experiment, follow-ups will be required to rule out associative learning explanations.

Another change we made was that we added "motivation" trials to the test sessions. These trials featured one prey item and no competitor; subjects could collect the prey every time. We elected to include these to avoid subjects' levels of frustration becoming too elevated as a result of frequently losing out on rewards in test and control trials. Each session contained 10 test trials, 10 control trials, and 4 motivation trials.

Please see Supporting S2 Video for a clip of subject Carola completing two test and one control trial.

## Analysis

As in Experiment 1, we conducted binomial tests to compare observed success to chance levels, and paired t-tests to compare Test and Control performance. In addition to conducting separate analyses for each subject, we also grouped subjects' data together and analyzed their behavior as a whole. All analysis was conducted in R version 4.1.2 [25].

## Results

Success levels for each subject across both conditions in the test are shown in Fig 7.

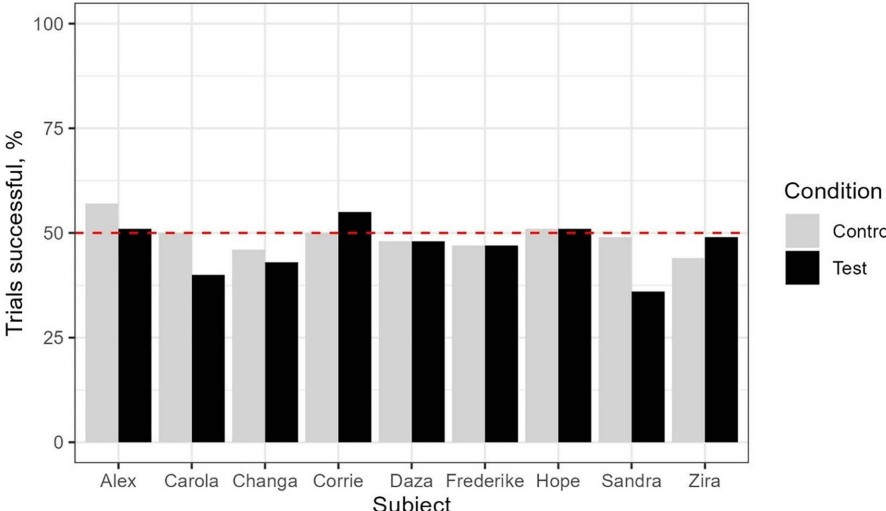

**Fig 7. Individual success rates across the control and test conditions in Experiment 2.** This figure indicates, for each subject, the percentage of trials in which they succeeded in both the Test and the Control conditions. The red dotted line shows the chance level of 50%.

For both conditions, binomial tests indicated that most subjects performed at chance levels. Only one subject, Sandra, differed from chance: she performed worse than expected in the Test (binomial test, p < .01) but not in the Control (binomial test, p = .920) (S4 Table in the Supporting Information).

In the t-tests, no significant difference was found between the Test and Control conditions for the group as a whole (t(8)=1.18, p = .273). Individual analyses revealed no significant differences between the Test and Control conditions for any of the subjects (Table 6), except for Sandra, who was significantly more successful in the Control than the Test (t(9)=2.86, p = .0187).

As in Experiment 1, we quantified all subjects' side biases and prey biases using binomial tests. Only four of the nine subjects displayed a side bias, all preferring to approach the left-hand prey (Supporting S5 Table). With regards to prey, results were similarly mixed: in Control trials, four subjects significantly preferred the chicken over the antelope, while

**Table 6. T-tests comparing individual and whole-group success rates across the control and test conditions.**

| Subject | t | Degrees of freedom | P |
|---|---|---|---|
| Alex | 1.032 | 9 | .329 |
| Carola | 0.410 | 9 | .691 |
| Changa | 0 | 9 | 1 |
| Corrie | −0.921 | 9 | .381 |
| Daza | 0.190 | 9 | .853 |
| Frederike | 0 | 9 | 1 |
| Hope | V = 10.5 *see note | NA | 1 |
| Sandra | 2.862 | 9 | .019 |
| Zira | −1.246 | 9 | .244 |
| All | 1.178 | 8 | .273 |

This table indicates the results of t-tests that were used to identify, at both individual and group levels, whether success rates differed significantly between the Test and Control conditions. The data from subject Hope failed to meet normality expectations, so these data were analyzed using a Wilcoxon signed-rank test rather than a parametric t-test.

three preferred the antelope. In the Test, four subjects preferred the rabbit and three preferred the boar (see Supporting S6 Table).

## Discussion

In Experiment 2, as in Experiment 1, we found no evidence for our chimpanzee subjects attributing preferences to the gorilla and elephant competitors and choosing their own actions accordingly. Experiment 2 was intended as a simplified version of the task, in which we attempted to address the possibility that the original setup, with regards to our efforts to preclude associative learning, prevented the chimpanzees from being able to understand the competitors' preferences. We approached this by adjusting the types of trials presented in the training stage so that subjects experienced more frequent successes *both* in terms of numbers of trials *and* in terms of numbers of rewards given, in the presence of a competitor and its non-preferred prey. Given that the results of Experiment 2 did not differ meaningfully from those of Experiment 1, this explanation appears implausible.

## General discussion

Neither of our experiments provided any evidence that chimpanzees attached stable preferences to competitors based on observing the type of prey that they took (and the prey that they did not). An obvious interpretation of these results is that chimpanzees cannot respond to others' preferences—possibly because they simply do not pay much attention to them, or because preferences that differ from their own create a meta-representational challenge, preventing chimpanzees from processing such information. This interpretation would be would be in line with a recent study [15] which found likewise that chimpanzees (and other great ape species) failed to strategically select food items based on the presence of different competitors with distinct preferences. However, previous research has suggested that in both cooperative and competitive [14] interactions, chimpanzees do take account of this information and use it to access food.

How do we explain these contradictory sets of results? One conspicuous difference between the current work and these prior studies [14,15] is the stimulus type: real human experimenters acting on real food items versus virtual animals competing for virtual targets that *represent* food rewards. It is possible that subjects in our study failed to interpret the stimuli in the desired way due to their virtual nature. Although investigations into their spatial cognition indicate that non-human primates behave similarly in virtual environments as in physical environments [40], much more work is required before we can confidently make assertions as to how they perceive and interpret on-screen stimuli, especially with respect to social interactions. Nevertheless, we feel it is clear that our subjects did at least understand the relationship between "collecting" one of the virtual prey targets and receiving a real fruit reward; and in part due to their exhibiting the expressions of frustration that we mentioned earlier, we believe the chimpanzees went at least some way towards interpreting the virtual gorilla and elephant as competitors. Moreover, chimpanzees in previous experiments have shown successful social learning from on-screen demonstrations [41,42], indicating that the absence of live stimuli is not the sole explanation for their failure in the current study. Additionally, even though the experiments in [15] used live human competitors, the chimpanzees still failed to act according to their preferences. This suggests that the use of virtual stimuli may not be the reason for negative results, since they can also occur with live stimuli.

As we touched on in the Introduction, another difference is timings: our experiments necessitated subjects making their choice in each trial *before* the competitor had acted; while the experiments in and [14] allowed chimpanzees to observe the actor carrying out an ambiguous action, and *then* tasked them with inferring which of two possible choices the actor had made. Perhaps it is less cognitively taxing and therefore easier for chimpanzees to consider retroactively what decision an individual might have made, having been afforded the opportunity to watch them making it, than it is to anticipate such an action.

However, another piece of work suggests that this is not the case. In an eye-tracking study, Kano and Call [31] found that chimpanzees, bonobos and orangutans predicted which of two targets a human hand would reach towards. In the

familiarization, subjects were shown videos of a human hand reaching for and grasping one of two target objects. The hand always selected the same object in every familiarization trial. The locations of the two objects were then switched. In the test, the hand began to move towards the objects but then stopped at an equal distance from both of them. The authors found that subjects preferentially looked towards the target object over the distractor, despite the fact that they had switched places, demonstrating that they anticipated that the hand would grasp the same object it had chosen previously. This result showed that the apes were conducting on-line goal-based predictions in a manner that is strikingly similar to what is seen in human infants in the first year of life [43]. It should be noted, however, that as this was an eye-tracking experiment, it did not require subjects to make any explicit decisions, instead capitalizing on their implicit expectations of the hand's actions.

There is one more difference that we would like to note between the current experiments and those from previous studies that showed successful preference attribution by chimpanzees. In our study, subjects never saw the competitors making an *overt* choice between the prey types. Subjects only ever saw one type of prey target present at a time across training trials, which the competitor either approached or ignored. Therefore, to succeed in the test, they were required to integrate competitors' behaviors from across disparate trials to develop the understanding that the competitor preferred one target over the other, rather than directly extrapolating from a single training trial to a single test. In contrast, the prior studies we have been discussing presented both options simultaneously. In Kano and Call [31], subjects always saw the hand overtly selecting between both targets in the familiarization trials; it may then have been straightforward to extend this selection to the test trials. Similarly, in the familiarization phase of Eckert et al., each experimenter drew their preferred piece of food from a population that contained both options. In Buttelmann et al. [14], in each trial, subjects saw the experimenter reacting to the contents of both boxes in quick succession. We therefore suggest that the added step of taking account of the competitor's behavior from trial to trial that was required in our experiment, in comparison to other studies which explicitly displayed the actor's preferences within the confines of a single trial, could be the cause of the differences in results.

Future work could clarify the relationship between these bodies of work in several ways. One possibility would be to replicate the current study with eye-tracking outcome measures. If the results mirror those of Kano and Call [31]—i.e., chimpanzees show first looks that skew preferentially towards the target object of each competitor—this would provide added evidence that chimpanzees are more likely to succeed at preference attribution tasks that involve implicit outcome measures than explicit choices. The use of these outcome measures should also help to reduce the impact of the subjects' side biases, as the behavioral lateralization and handedness discussed earlier is less likely to affect looking time and direction. Such an experiment could additionally compare chimpanzees' performance in a live test versus a virtual one, and thereby investigate whether they do better at comprehending competitors' preferences when these competitors are live human experimenters with whom they can interact in various ways, as opposed to virtual animals.

Another interesting avenue might be to replicate the current study and keep the same outcome measures but change the format of the training so that a competitor is shown choosing between both targets on each trial. For example, training trials could comprise the dog prey on one side and the rabbit on the other, with the gorilla always selecting the dog and the elephant always going for the rabbit. The prey would then switch sides, and test trials would be the same as the current experiment: depending on which competitor is present, the subject can choose which prey to approach. If the chimpanzees were to show improved success in this format in comparison to the results presented here, the above conclusion, that preferences are easier to comprehend when they are overt versus implicit, would be strengthened.

## Conclusion

Our two experiments investigated chimpanzees' ability to attribute preferences to virtual competitors and to adjust their behaviors accordingly. We hypothesized that chimpanzees would respond to competitors' preferences and select the correct prey at above-chance levels in the Test, but not in the Control.

The results of Experiment 1 did not support this hypothesis. In Experiment 2, we investigated the likelihood that the original results were due to our potentially overly complex and demanding training scheme. However, the results mirrored those of Experiment 1.

We have posited several other explanations for chimpanzees' failure in this test—insufficient saliency of stimuli; low stakes; having to rely on visual information alone; side and stimulus biases; timing of decision-making; presence of only one target per trial in the training phase. Drawing upon related studies that have shown positive results, we have suggested follow-up experiments which we believe have the potential to rule out or confirm some of these alternatives, such as running the same experiment again as an eye-tracking test, or adjusting the training such that the competitors' preferences are more explicitly displayed.

## Supporting information

**S1 Table. Experiment 1 observed success rates and binomial tests.**
(DOCX)

**S2 Table. Experiment 1 individual subject side biases (Binomial tests).**
(DOCX)

**S3 Table. Experiment 1 individual subject prey biases (Binomial tests).**
(DOCX)

**S4 Table. Experiment 2 observed success rates and binomial tests.**
(DOCX)

**S5 Table. Experiment 2 individual subject side biases (Binomial tests).**
(DOCX)

**S6 Table. Experiment 2 individual subject prey biases (Binomial tests).**
(DOCX)

**S1 Video. Clips of Azibo completing trials in the test phase of Experiment 1.**
(MP4)

**S2 Video. Clips of Carola completing trials in the test phase of Experiment 2.**
(MP4)

## Acknowledgments

We extend our thanks to the Wolfgang Köhler Primate Research Centre at Leipzig Zoo, where we collected the data for this study. We thank the zookeepers and veterinary team for their care of the chimpanzees and their facilitation of this research. We thank Hanna Petschauer and Sebastian Schütte for their coordination of research at the WKPRC, and Cristobal Cantuarias for his help with data collection.

## Author contributions

**Conceptualization:** Emilie Rapport Munro, Matthias Allritz, Josep Call.

**Data curation:** Emilie Rapport Munro.

**Formal analysis:** Emilie Rapport Munro.

**Investigation:** Emilie Rapport Munro.

**Methodology:** Emilie Rapport Munro, Matthias Allritz, Josep Call.

**Project administration:** Emilie Rapport Munro, Daniel B.M. Haun.

**Resources:** Daniel B.M. Haun.

**Software:** Kenneth Schweller.

**Supervision:** Matthias Allritz, Josep Call.

**Validation:** Emilie Rapport Munro.

**Visualization:** Emilie Rapport Munro.

**Writing – original draft:** Emilie Rapport Munro.

**Writing – review & editing:** Emilie Rapport Munro, Matthias Allritz, Daniel B.M. Haun, Josep Call.

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
