## [Decision Letter · Decision Letter 0]

17 May 2025

PONE-D-24-58999Do chimpanzees (Pan troglodytes) attribute preferences to virtual competitors?PLOS ONE?

Dear Dr. Rapport Munro,

Thank you for submitting your manuscript to PLOS ONE. After careful consideration, we feel that it has merit but does not fully meet PLOS ONE’s publication criteria as it currently stands. Therefore, we invite you to submit a revised version of the manuscript that addresses the points raised during the review process.

We look forward to receiving your revised manuscript.

Kind regards,

James Edward Brereton, MSc

Academic Editor

PLOS ONE

Journal Requirements:

4. Please remove all personal information, ensure that the data shared are in accordance with participant consent, and re-upload a fully anonymized data set.

Reviewers' comments:

Reviewer's Responses to Questions

**Comments to the Author**

1. Is the manuscript technically sound, and do the data support the conclusions?

Reviewer #1: Partly

Reviewer #2: Yes

2. Has the statistical analysis been performed appropriately and rigorously?

Reviewer #1: No

Reviewer #2: Yes

3. Have the authors made all data underlying the findings in their manuscript fully available?

Reviewer #1: No

Reviewer #2: Yes

4. Is the manuscript presented in an intelligible fashion and written in standard English?

Reviewer #1: Yes

Reviewer #2: Yes

Reviewer #1: This study investigated the ability of chimpanzees to attribute preferences to virtual competitors and adjust their behavior accordingly. Two experiments were conducted, but no statistically significant results were found regarding the use of memory about competitors' preferences. Some behavioral aspects inherent to social animals raised concerns, as there could be associative learning where chimpanzees adjust their responses based on previous trials. To control for this, the study included control conditions where no information about competitors' preferences was provided. The results showed no significant differences between the test and control conditions, suggesting that chimpanzees were not using preference information to guide their choices.

Major Comments:

The introduction is excessively long and becomes tedious due to unnecessary detail. It could be improved by summarizing all examples or focusing only on the discrepancies—below, I provide some examples of how to condense it. While I understand the purpose of the study, I suggest that, given the extensive context and discrepancies regarding these behavioral aspects, the authors should clearly define a couple of specific objectives and formulate explicit hypotheses.

I find the procedure difficult to follow. Perhaps the authors could add a graphical representation of the procedure, explaining the stages and specifying the number of trials in each experiment so that the reader can properly follow the procedure and understand the number of trials, experimental units, and test runs.

My main concern is that the structure of the article does not align with PLOS ONE’s formatting. The authors present the Methods, Results, and Discussion sections separately for each experiment. I disagree with this approach because it becomes repetitive. The authors should restructure the manuscript so that I can better focus on the study's scientific contributions.

Specific Comments:

• Lines 60-80: Specify which of these studies involved experiments with virtual animals or clarify the conditions in which they were conducted.

• Lines 44-60: This paragraph provides context about multilevel societies in chimpanzees and the importance of avoiding competition. It could be condensed by removing specific examples and focusing on the main idea.

• Lines 61-66: This paragraph provides an example of a previous study on chimpanzees. Consider summarizing it, keeping only the key idea and removing unnecessary study details.

• Lines 67-72: This paragraph describes chimpanzees' behavior in competitive situations. It could be synthesized to focus on the relevance of understanding competitors' preferences without including excessive examples.

• Lines 81-102: This section provides background on competition in chimpanzees and the importance of understanding how they interpret others' preferences. It could be condensed to highlight the relevance of the research without excessive details on previous studies.

• Lines 103-148: This section analyzes prior research on primate competitive behavior. Summarizing key findings while mentioning only the most relevant results would help clarify the purpose of this study.

• Lines 161-191: This section justifies the chosen experimental approach. The justifications could be condensed, focusing on the main rationale behind the design choices and eliminating redundancies.

• Lines 192-204: The authors provide excessive details that belong in the Methods section—remove them. This section should include a couple of specific objectives (one for each experiment) and the hypotheses to be tested, given the extensive background provided.

Methods:

• Line 207: The Methods section should be placed first.

• Line 228: Specify the number of trials per individual.

• Many paragraphs are too short and disconnected, e.g., lines 278-281 and 308-310—merge them with the following paragraphs.

• Table 1: I do not understand this table. A diagram of the procedure with images and connecting lines illustrating the experiment’s structure would be more effective.

• Line 432: Software should be mentioned at the end of the section, along with references for any statistical packages used.

• Line 439: The authors introduce hypotheses here—why were they not stated in the introduction?

• Line 444: This paragraph stands alone without context. Specify the statistical packages used and cite them. Also, indicate whether the data met normality assumptions and explain why a paired t-test was used—while it is known that paired tests are for related samples, the reader needs clarification. Explain what test was used when the data were non-normal. Additionally, in the binomial test, specify what constitutes success and what constitutes failure, as this distinction affects the formula's properties and the hypothesis test. Further justification for the 50% assumption in the formula is needed—this seems arbitrary, and references should support it. Otherwise, a χ² goodness-of-fit test might be more appropriate for frequency data.

Reviewer #2: Dear authors,

First, I would like to congratulate you on the excellent work. This is a highly complex study, with a carefully designed methodology, analyses that are simple yet robust enough to test the hypotheses, and an outstanding discussion.

That said, I have made a few comments on the manuscript, which are detailed below.

First, some general comments.

Although the text flows well, it is relatively long—especially the introduction. I believe certain parts could be shortened or moved to the Methods section. This would make the reading experience more pleasant.

Another important point is that the figure and table captions are overly simplistic. Captions need to be self-explanatory, and in their current form, they are not. I therefore recommend revising them to ensure they meet this criterion.

It would be helpful if, in both experiments, the description of the animals included tables with their individual information. This is important because throughout the text, the individuals are referred to by name, which can be difficult for readers to follow. Tables including names, sex, ages, experience with virtual games, etc., would make this much clearer.

Another point of interest concerns how the chimpanzees behaved during the tests. Were they consistently engaged in the task, or only at the beginning? Did they lose interest after failing a trial? Did they pay attention to the screen or were they more focused on their surroundings?

Videos of the test sessions would make excellent supplementary material. I recommend including a few clips if possible.

Another question concerns the effectiveness of the rewards (fruit pieces). You mention that the animals were not food-deprived for the tests, but were the tests conducted close to feeding times? Do you think the animals might have lacked motivation for the rewards, which could explain a lack of effort in trying to obtain them?

Finally, do you think individual differences in personality or cognitive ability may have influenced the results? I believe these aspects would be interesting to discuss in the manuscript.

Additional minor suggestions:

Line 49: Insert references after the word "coalitions."

Lines 211–214: Delete the part about the individual who did not participate in the experiment, as this information is irrelevant. This would also help reduce the length of the text.

Line 224: Please provide the protocol numbers for the ethics committee approvals.

Lines 275–277: In a trial with four prey items, if the chimpanzee captured only one of the four, would they still receive the reward or not? Please clarify.

Lines 300–306: Delete this information.

Line 359: Replace "3" with "three."

Line 373: Replace "9" with "nine."

Line 382: Delete the period after "trials."

Lines 490–496: This paragraph is very good. I actually had the same thought — could the chimpanzees have learned how to play the game? Would you happen to have data from the training sessions to assess whether their responses changed over time?

Line 540: Do you mean "physical alteration" here? Please clarify.

Lines 562–601: I was wondering whether the chimpanzees may have learned that losing in this game doesn't really matter. It's similar to when we play a game passively and don’t care what happens to our avatar. But I’m not sure this applies here, since some animals actually appeared frustrated by their losses. You also discuss the relevance of virtual versus real-world settings, noting that real situations might have even physical consequences. I really appreciated this discussion.

**Do you want your identity to be public for this peer review?** For information about this choice, including consent withdrawal, please see our Privacy Policy

Reviewer #1: **Yes: ** John F. Aristizabal

Reviewer #2: No

---

## [Author Response · Author response to Decision Letter 1]

18 Jun 2025

Reviewer #1

This study investigated the ability of chimpanzees to attribute preferences to virtual competitors and adjust their behavior accordingly. Two experiments were conducted, but no statistically significant results were found regarding the use of memory about competitors' preferences. Some behavioral aspects inherent to social animals raised concerns, as there could be associative learning where chimpanzees adjust their responses based on previous trials. To control for this, the study included control conditions where no information about competitors' preferences was provided. The results showed no significant differences between the test and control conditions, suggesting that chimpanzees were not using preference information to guide their choices.

We thank you for your constructive suggestions, which have helped us to render the manuscript clearer and more concise.

Major Comments:

1. The introduction is excessively long and becomes tedious due to unnecessary detail. It could be improved by summarizing all examples or focusing only on the discrepancies—below, I provide some examples of how to condense it. While I understand the purpose of the study, I suggest that, given the extensive context and discrepancies regarding these behavioral aspects, the authors should clearly define a couple of specific objectives and formulate explicit hypotheses. We have condensed the introduction and added hypotheses to it; for specific changes, please refer to our responses to comments #4-11.

2. I find the procedure difficult to follow. Perhaps the authors could add a graphical representation of the procedure, explaining the stages and specifying the number of trials in each experiment so that the reader can properly follow the procedure and understand the number of trials, experimental units, and test runs. We have created a graphic for each experiment to show the trial types in each phase of the procedure, how many trials were of each type, and how many rewards subjects would receive in each trial. These are Figs 2 and 5.

3. My main concern is that the structure of the article does not align with PLOS ONE’s formatting. The authors present the Methods, Results, and Discussion sections separately for each experiment. I disagree with this approach because it becomes repetitive. The authors should restructure the manuscript so that I can better focus on the study's scientific contributions. We appreciate your feedback on this point, but we are concerned that this change would render the manuscript harder to follow, not easier. Our reasoning for structuring the manuscript in the current form is that the rationale and design for Experiment 2 are entirely predicated upon the Results and Discussion of Experiment 1. We feel it would be harder for the reader to understand our reasons for running Experiment 2 in the way we did if we introduced its methodology before discussing the results of Experiment 1. To avoid repetitiveness, in instances where the methodology of Experiment 2 was the same as Experiment 1, we have stated that, rather than repeating the details. We have considered this carefully, and have sought the advice of the editor, who has agreed that it is acceptable for the structure of the manuscript to be left as we originally wrote it.

Specific Comments:

4. Lines 60-80: Specify which of these studies involved experiments with virtual animals or clarify the conditions in which they were conducted. None of these studies included virtual animals. We have adjusted the text to clarify the conditions:

Lines 55-59: Hare et al. (7) conducted an experiment in which a subordinate and a dominant chimpanzee entered the same room from different sides. The room contained two pieces of food: one was visible to both chimpanzees, while the other was visible to the subordinate but screened from the dominant. Subordinate chimpanzees chose to approach the item that was hidden from the dominant’s view.

5. Lines 44-60: This paragraph provides context about multilevel societies in chimpanzees and the importance of avoiding competition. It could be condensed by removing specific examples and focusing on the main idea. We have removed the examples in the first paragraph concerned, but we have left the second paragraph as we feel it contains important context setting up the experiments discussed in the next paragraphs.

6. Lines 61-66: This paragraph provides an example of a previous study on chimpanzees. Consider summarizing it, keeping only the key idea and removing unnecessary study details. We have not been able to summarise this paragraph as doing so would conflict with comment #4, which requested additional details on the study’s methodology.

7. Lines 67-72: This paragraph describes chimpanzees' behavior in competitive situations. It could be synthesized to focus on the relevance of understanding competitors' preferences without including excessive examples. We have shortened this paragraph. It now reads:

Lines 62-65: An initial replication attempt failed to find similar evidence of perspective-taking (8), but Bräuer and colleagues later succeeded in replicating the original results, noting that the first replication suffered from a lack of statistical power and from methodological issues (9).

8. Lines 81-102: This section provides background on competition in chimpanzees and the importance of understanding how they interpret others' preferences. It could be condensed to highlight the relevance of the research without excessive details on previous studies. We have condensed this section to remove specific examples.

Lines 65-72: Note that although critics of mentalistic accounts of apparent perspective-taking in chimpanzees have put forward alternative explanations, such as behavior-reading and stimulus generalization from prior experience, both the Hare and Bräuer studies included experimental conditions that controlled for these explanations. Various other food competition paradigms have accordingly suggested visual-perspective-taking abilities in chimpanzees: they conceal their approaches from competitors (10), intentionally deceive competitors by approaching food indirectly (11), and strategically manipulate competitors’ visual access (12).

9. Lines 103-148: This section analyzes prior research on primate competitive behavior. Summarizing key findings while mentioning only the most relevant results would help clarify the purpose of this study. This section has been shortened, with extra details removed, but we have also added a reference to another recent study because it is very relevant and provides contrasting results to the other studies discussed. Lines 80-102:

Eckert and colleagues (13) have shed some light on this question. In their study, chimpanzees gained experience of the conflicting food biases of two human experimenters; then, in test trials, the chimpanzees expected the experimenters to make choices in line with their biases, except when they could not see what they were choosing.

In another study, Buttelmann and colleagues (14) demonstrated that chimpanzees inferred which of two boxes a human experimenters would have eaten from based on the valence and directedness of their emotional expressions. The chimpanzees therefore chose to search for food in the other, untouched box.

Recently, Kaminski and colleagues (15) found contrasting results. In their study, subjects chose between two food items after a human competitor had made a choice in secret, such that the subjects did not know which item was still available. When interacting with a competitor whose biases matched the subjects’, the correct response was to choose their own less preferred food. All four species of great ape failed to do so. This could indicate that the apes were unable to recognize the competitors’ preferences; however, another potential explanation is that they were merely unable to inhibit their own desire to select their preferred food. Considering the conflicting findings of the above studies, further work is required in a variety of contexts to provide robust support or contradictions of the hypothesis that chimpanzees recognize to others’ preferences. The current study aims to deliver this, using a novel method for simulating social interactions: competing with virtual agents.

10. Lines 161-191: This section justifies the chosen experimental approach. The justifications could be condensed, focusing on the main rationale behind the design choices and eliminating redundancies. We have condensed this section from 30 lines to 17.

Lines 102-119: Over the past several years, virtual environments have been used with increasing frequency to study great apes’ navigational and spatio-cognitive skills (16–21). Rapport Munro and colleagues (20) recently conducted the first experiment to investigate social cognition within the virtual realm, in which chimpanzees and bonobos were tasked with “catching” (i.e. guiding their invisible virtual avatar to collide with) virtual rabbits that ran away when approached. All subjects learned rapidly how to chase and catch the rabbits, despite only previously having experience with static virtual targets; moreover, generative computational modeling techniques revealed that subjects occasionally intercepted the rabbits by anticipating their movement trajectories. The benefit of using virtual environments to study ape social cognition is that they provide a good balance between logistical and ethical considerations on the one hand, and experimental validity on the other hand. Some of the best and most-cited evidence of chimpanzees considering the states of minds of others (7,22) comes from real-life competition experiments involving two conspecifics simultaneously going after the same target in the same room. However, important ethical and welfare advances have rendered such experiments difficult to administer. Using virtual environments, we can return to something akin to those original methods while also controling for potential behavior-reading and prior-experience explanations.

11. Lines 192-204: The authors provide excessive details that belong in the Methods section—remove them. This section should include a couple of specific objectives (one for each experiment) and the hypotheses to be tested, given the extensive background provided. We have shortened this section and moved the hypotheses from the methods section to here. Lines 120-127: The current study investigated whether chimpanzees could learn the differing prey preferences of two novel virtual competitors, and subsequently use this knowledge to succeed in competitive interactions by choosing to approach the non-preferred prey of whichever competitor was present. To rule out associative learning explanations, we also included a control condition featuring prey items with regards to which subjects had no information about the competitors’ potential biases. We hypothesized that if the chimpanzees were responding to the competitors’ preferences, they should choose the correct prey at above-chance levels in the Test condition, but not in the Control.

Methods:

12. Line 207: The Methods section should be placed first. We apologise, but we do not quite understand this comment. The Methods section cannot be placed any earlier, as it needs to follow the introduction.

13. Line 228: Specify the number of trials per individual. Line 228 in the original document and the subsequent lines refer to the Materials used in Experiment 1; we do not believe that this is a suitable place to state the number of trials per individual. Under ‘Procedure’, we specify how many trials each individual received in each stage of the experiment; we have also added, in the ethics section, specifics on the maximum number of trials a subject could receive per day, alongside the maximum duration of testing times.

Lines 142-143: Except for dependent infants, each subject was separated from the rest of their group for the testing period, which never exceeded 30 minutes per subject per day (with a maximum of 24 trials per day).

14. Many paragraphs are too short and disconnected, e.g., lines 278-281 and 308-310—merge them with the following paragraphs. We have merged these paragraphs.

15. Table 1: I do not understand this table. A diagram of the procedure with images and connecting lines illustrating the experiment’s structure would be more effective. We have created a graphical representation of the procedure for this and for Experiment 2.

16. Line 432: Software should be mentioned at the end of the section, along with references for any statistical packages used. We have clarified that only base R was used.

17. Line 439: The authors introduce hypotheses here—why were they not stated in the introduction? We have moved the hypotheses to the Introduction (see also our response to comment #11).

18. Line 444: This paragraph stands alone without context. Specify the statistical packages used and cite them. Also, indicate whether the data met normality assumptions and explain why a paired t-test was used—while it is known that paired tests are for related samples, the reader needs clarification. Explain what test was used when the data were non-normal. Additionally, in the binomial test, specify what constitutes success and what constitutes failure, as this distinction affects the formula's properties and the hypothesis test. Further justification for the 50% assumption in the formula is needed—this seems arbitrary, and references should support it. Otherwise, a χ² goodness-of-fit test might be more appropriate for frequency data. We have made some changes to this section. We have clarified why a chance level of 0.5 was used and have added a reference for this being standard practice. We have defined success and failure. We have clarified that base R was used, with no additional packages. We have explained why we used paired-samples t-tests.

Lines 373-393: First, we wished to ascertain whether the observed success levels differed from those expected by chance. We set the chance level at 0.5, as is standard for binomial tests in which under the null hypothesis (i.e. randomness) the expected probability of success on any one trial is 0.5 (26). In our study, there were two choice options (two prey items) in every trial, and “success” was denoted as where a subject chose correctly, i.e. chose the non-preferred prey of the competitor present, while “failure” was where they chose incorrectly. We anticipated, when planning the test, that the percentage of trials that passed without one of these two choices being made would be negligible based on the subjects’ high success rate in the training stages in which prey could be caught. This informed our decision to set the chance level of 0.5. Indeed, there were zero instances of a chimpanzee not choosing one of those options (i.e., choosing not to “move” at all within the virtual environment, or “walking” in a direction not towards either of the prey). We conducted separate binomial tests for the Test condition and the Control condition.

Second, we compared performance between Test and Control conditions. We assessed the normality of data using a Shapiro test. If the normality assumption was not violated, we used a parametric paired samples t-test, where “session” served as the grouping variable. The reason for using paired samples t-tests was that the samples were non-independent due to the within-subjects design. If the data were non-normal, we used a Wilcoxon signed-rank test instead.

All analysis was conducted in R version 4.1.2 (25). Base R was used, with no additional packages employed.

Reviewer #2

Dear authors,

First, I would like to congratulate you on the excellent work. This is a highly complex study, with a carefully designed methodology, analyses that are simple yet robust enough to test the hypotheses, and an outstanding discussion. Thank you very much for your comments. We are particularly pleased that you feel we covered all areas of interest in the Discussion.

That said, I have made a few comments on the manuscript, which are detailed below.

First, some general comments.

19. Although the text flows well, it is relatively long—especially the introduction. I believe certain parts could be shortened or moved to the Methods section. Thi

---

## [Decision Letter · Decision Letter 1]

17 Jul 2025

Do chimpanzees (*Pan troglodytes* ) attribute preferences to virtual competitors?

PONE-D-24-58999R1

Dear Dr. Rapport Munro,

We’re pleased to inform you that your manuscript has been judged scientifically suitable for publication and will be formally accepted for publication once it meets all outstanding technical requirements.

Kind regards,

James Edward Brereton, MSc

Academic Editor

PLOS ONE

Additional Editor Comments (optional):

Reviewers' comments:

Reviewer's Responses to Questions

**Comments to the Author**

Reviewer #1: All comments have been addressed

Reviewer #2: All comments have been addressed

2. Is the manuscript technically sound, and do the data support the conclusions?

Reviewer #1: Yes

Reviewer #2: Yes

3. Has the statistical analysis been performed appropriately and rigorously?

Reviewer #1: Yes

Reviewer #2: Yes

4. Have the authors made all data underlying the findings in their manuscript fully available?

Reviewer #1: Yes

Reviewer #2: Yes

5. Is the manuscript presented in an intelligible fashion and written in standard English?

Reviewer #1: Yes

Reviewer #2: Yes

Reviewer #1: This study investigated the ability of chimpanzees to attribute preferences to virtual competitors and adjust their behavior accordingly. Two experiments were conducted, but no statistically significant results were found regarding the use of memory about competitors' preferences. The study provides valuable insights into the limitations and challenges in interpreting their social cognition. Additionally, the authors demonstrated significant effort in improving the experimental design, carefully attending to all our suggestions with great dedication. Overall, this work represents an important contribution to understanding chimpanzee cognitive abilities, showcasing methodological rigor and an openness to future research in this field. I have a minor comments, considering them, an additional revision is not required.

Minor

Line 86–89: Merge this paragraph with the previous one.

Table 1: Explain what “VR” means in the table title.

Line 407: P should be uppercase and italicized; include the degrees of freedom.

Line 452: I understand that “real world” refers to real-life conditions as opposed to virtual ones, but it would be helpful for readers if this were specified in parentheses.

Line 573: It’s good that it doesn’t sound repetitive—this was my suggestion—but I can’t imagine reading such a simplistic sentence in a scientific article. I suggest explicitly stating which part of the experiment is the same.

Line 648: Move the sentence “All analysis was conducted in R version 4.1.2 (25).” to the end of the paragraph.

Line 665: Does “(8)” refer to the degrees of freedom? P should be uppercase and italicized. Please check the rest as well.

Line 800: Explicitly state whether the study supported the hypothesis or not.

The figures are of very poor quality, though this is likely due to the proofing process.

The videos are wonderful.

Reviewer #2: Dear authors,

Thank you for submitting the revised version of your manuscript.

This version is more concise and focused, which makes for a much more pleasant read. Furthermore, the new figures and, most notably, the videos provide a much clearer understanding of the experiment. Well done on that.

In my view, the manuscript is now acceptable, but I have a few minor corrections (typos) to suggest, which do not necessitate another round of revisions:

Line 97: Please move the full stop after (15).

Line 159: Please delete the full stop after (23) and keep the phrase in parentheses following it, starting with a lowercase letter (we....).

Line 314: Please change '6' to 'six'.

Line 319: Please change '1' to 'one'.

Those are my comments. Congratulations once again on your very interesting study.

**Do you want your identity to be public for this peer review?** For information about this choice, including consent withdrawal, please see our Privacy Policy

Reviewer #1: No

Reviewer #2: No

---

## [Editor Report · Acceptance letter]

PONE-D-24-58999R1

PLOS ONE

Dear Dr. Rapport Munro,

I'm pleased to inform you that your manuscript has been deemed suitable for publication in PLOS ONE. Congratulations! Your manuscript is now being handed over to our production team.

Kind regards,

on behalf of

Mr. James Edward Brereton

Academic Editor

PLOS ONE